# Nuphar lutea Extracts Exhibit Anti-Viral Activity against the Measles Virus

**DOI:** 10.3390/molecules25071657

**Published:** 2020-04-03

**Authors:** Hila Winer, Janet Ozer, Yonat Shemer, Irit Reichenstein, Brit Eilam-Frenkel, Daniel Benharroch, Avi Golan-Goldhirsh, Jacob Gopas

**Affiliations:** 1Department of Microbiology and Immunology and Genetics, Faculty of Health Sciences, Ben-Gurion University of the Negev, Beer Sheva 8400501, Israel; hilawiner@gmail.com (H.W.); yonat@bgu.ac.il (Y.S.); iritreic@gmail.com (I.R.); brit1812@gmail.com (B.E.-F.); jacob@bgu.ac.il (J.G.); 2Department of Clinical Virology, Faculty of Health Sciences, Ben-Gurion University of the Negev, Beer Sheva 8400501, Israel; 3Department of Pathology, Faculty of Health Sciences, Ben-Gurion University of the Negev, Beer Sheva 8400501, Israel; benaroch@bgu.ac.il; 4The Jacob Blaustein Institutes for Desert Research; Albert Katz Department of Dryland Biotechnologies, Sede Boqer, Ben-Gurion University of the Negev, Beer Sheva 8400501, Israel; avigolan@bgu.ac.il; 5Department of Oncology, Soroka University Medical Center, Beer Sheva 8400501, Israel

**Keywords:** herbal medicines, *Nuphar lutea* plant extract, nupharidines, measles virus, anti-viral natural products

## Abstract

Different parts of *Nuphar lutea L*. (yellow water lily) have been used to treat several inflammatory and pathogen-related diseases. It has shown that *Nuphar lutea* extracts (NUP) are active against various pathogens including bacteria, fungi, and leishmanial parasites. In an effort to detect novel therapeutic agents against negative-stranded RNA (- RNA) viruses, we have tested the effect of a partially-purified alkaloid mixture of Nuphar lutea leaves on the measles virus (MV). The MV vaccine’s Edmonston strain was used to acutely or persistently infect cells. The levels of several MV proteins were detected by a Western blot and immunocytochemistry. Viral RNAs were quantitated by qRT-PCR. Virus infectivity was monitored by infecting African green monkey kidney VERO cells’ monolayers. We showed that NUP protected cells from acute infection. Decreases in the MV P-, N-, and V-proteins were observed in persistently infected cells and the amount of infective virus released was reduced as compared to untreated cells. By examining viral RNAs, we suggest that NUP acts at the post-transcriptional level. We conclude, as a proof of concept, that NUP has anti-viral therapeutic activity against the MV. Future studies will determine the mechanism of action and the effect of NUP on other related viruses.

## 1. Introduction

Natural products of *Nuphar lutea* (L.) SM. (Nymphaeaceae) have been widely used for treating inflammatory conditions in ethnic medicine.

In Lebanon, leaf extracts were used against rheumatism [1]. In Japan, a herbal mixture that includes *Nuphar* rhizome powder was used to treat swelling and pain in Jidabokuippo traditional medicine [2]. The local Gitksans of Northwestern British Columbia utilized *Nuphar polysepalum* to treat tuberculosis, fractures, arthritis, and other diseases [3]. The use of *Nupar lutea* extracts for medicinal purposes by aboriginals of the Canadian boreal forest was reported by Uprety et al. [4]. A systematic review of early studies revealed that the full therapeutic potential of *Nuphar* products is still largely unexplored by modern research [5]. Nevertheless, recent reports from various laboratories, including ours, on the medicinal properties of *Nuphar* extracts have indicated potential applications: as anti-inflammatory [6], anti-bacterial [7,8] against pathogenic fungi [9], against the leishmania and trypanosome parasites, [10,11,12], and against cancer [7,13,14].

The global impact of viral infections, the development of antiviral drug resistance, and the emergence of new viruses are all driving the incessant search for new compounds endowed with safe and effective antiviral activity.

In this context, natural products originating from botanical, animal, or mineral sources traditionally used in ethno-medicine may provide leads for modern antiviral drug development once their pharmacological potential is verified by scientific investigation.

The search for novel anti-viral agents has led to the identification of new molecular viral targets such as proteins involved in genome replication or in viral maturation and egress. Moreover, new strategies based on the identification of specific cellular targets required for viral replication have been developed. In this work, we describe the anti-viral properties of an alkaloid mixture extracted from *Nuphar lutea* (NUP). The major components of the partially purified mixture found by NMR analysis are dimeric sesquiterpene thioalkaloids called thiobinupharidines and thiobinuphlutidines [15].

The measles virus (MV) belongs to the *Paramyxoviridae* family. Most paramixoviruses are highly contagious airborne pathogens that spread via the respiratory route. Despite the presence of vaccination, there is exceptionally high infectivity of MV. Herd immunity of >95% is required to suppress endemic transmission. Due to low vaccination coverage in parts of the developing world and insufficient or declining herd immunity in several developed countries, MV still remains the leading cause of global childhood death from a vaccine-preventable disease [16]. Recently, MV-induced encephalitis among human immunodeficiency virus (HIV)-infected children has become a great concern in high–HIV-prevalent countries [17,18].

Thus, no therapeutic strategy is available to rapidly control local outbreaks or improve case management of severe measles. Further research must be done to find new treatments against this and other related viruses [19,20,21]. In this work, we demonstrated the in vitro anti-viral effect of NUP on MV acutely and persistently infected cells.

## 2. Results

### 2.1. NUP Protects Cells from Acute Cytotoxic Infection with MV

To determine whether *Nuphar lutea* (NUP) can protect cells from the cytotoxic effect of acute viral infection, we treated cells with non-toxic concentrations of NUP either before infection, simultaneously, or after infection with MV. For controls, we used African green monkey kidney VERO cells incubated only with medium, vehicle (50% methanol), or NUP. All the controls showed 100% survival and the efficiency of MV infection was not affected by treatment with the vehicle (data not shown). The cytotoxic effect of MV acute infection was examined by the Tetrazolium-based cell proliferation (viability) assay kit. The results show that NUP protected infected cells either when pretreated, added simultaneously, and after infection (Figure 1). The best results were achieved when cells were pretreated with NUP.

### 2.2. NUP Treatment Reduces the Amount of Released Viral Particles

To quantify the amount of virus released to the supernatant by NUP-treated and non-treated cells, a similar experiment was designed with VERO-SLAM cells infected with the MV wild type GFP labeled IC323-GFP strain (MV-GFP). 96 h post infection supernatants were collected and titrated as described in Materials and Methods. We observed a decrease by 10^4^-fold in virus concentration in cells pre-treated with NUP as compared to the untreated control. Cells initially infected with MV and then treated with NUP showed a 10^3^-fold viral decrease (Table 1).

### 2.3. NUP Treatment of L428 MV- Persistently Infected Cells Reduces the Amount Of Viral Protein

Since NUP treatment significantly reduces the cytotoxic effect of MV (Figure 1) as well as the amount of MV released to the supernatant of infected cells (Table 1), we next asked whether NUP affects the expression of viral proteins in the persistently infected cells. To this end, we generated MV-GFP persistently infected L428 cells. All cells in the culture expressed MV-GFP (Figure 2).

Following NUP treatment, viral proteins expression was detected by immunohistochemistry (Figure 3) and by a Western blot (Figure 4). We observed a dose-dependent decrease in MV-N and -V proteins expression and a complete inhibition in MV-P protein expression upon NUP treatment (Figure 4).

### 2.4. MV RNA Expression is not Affected by NUP Treatment

Since NUP reduces the expression of the MV viral proteins tested, we asked whether or not the reduction in MV protein amounts is a result of a decrease in viral RNA. To this end, we incubated L428 + MV cells for 12 h with a different concentration of NUP, isolated total RNA, and performed qRT-PCR on the MV N- and P- genes. We normalized the results to human β-actin and PHP genes as compared to untreated L428 + MV cells. The final results were calculated as the difference between the number of cycles (Ct) in control L428 + MV cells’ cycle and each of the samples treated with NUP (∆Ct). The highest difference observed between the treated and untreated samples was of 1.5 cycles (Figure 5). These results are not significant and suggest that NUP reduces the amount of MV viral proteins by a post-transcriptional rather than a transcriptional mechanism.

## 3. Discussion

A highly effective vaccine is available against the MV. Its prudent use could result in the eradication of MV infections whenever appropriate immunization strategies are used. Nevertheless, the MV still causes substantial morbidity and mortality throughout the world despite being a preventable infectious disease. Therefore, there is an urgent need to develop effective anti-viral therapies for non-immunized and individuals at high risk [20,22]. In this work, we have studied the in vitro anti-viral effect of NUP in acutely and persistently MV-infected cells.

The main obstacle in most studies searching for anti-viral agents is their cytotoxicity. NUP’s cytotoxicity is cell line dependent [15]. Therefore, we determined the range of NUP concentrations, which were not cytotoxic to L428 and VERO cells (data not shown). Non-toxic concentrations were used to investigate its anti-viral effect. Infected and non-infected counterpart cell lines showed similar sensitivity to NUP, which indicates that persistent viral infection *per se* does not change the sensitivity of the cells to NUP.

The MV is a lytic virus, and usually induces cell death in vitro after three to four days. Following infection, cytopathic, morphological changes are induced by the viruses. NUP treatment, especially NUP pretreatment of cells, enabled protection from MV acute infection (Figure 1).

Most existing anti-viral agents target viral proteins, including the MV proteins that are responsible for the virus’s fusion (F-protein), entry (H-, G-proteins), or replication (N-, L-proteins) [19,23,24,25,26,27]. Therefore, we studied whether NUP inhibits the expression of several viral proteins. We observed a decrease in MV N- and V-proteins and complete inhibition of P- protein expression (Figure 4). The N-reduction and P-reduction were confirmed by immunostaining (Figure 3).

The P-protein is a component of the RNA dependent RNA polymerase complex essential for viral replication [17] and may serve as a novel anti-viral target. Down regulation of this protein should efficiently inhibit viral replication, which results in protection from acute infection.

As shown by qRT-PCR, the effect of NUP is not mediated by MV viral RNA degradation, but rather by post-transcriptional mechanisms. Possible MV targets for therapeutic intervention that have been described, such as inhibition of replication by purine RNA nucleotides analogs like Ribavirin, which has limited clinically use, or anti-sense molecules, which target the N-proteins and H-proteins (studied only *in vitro*). Therapeutic agents that inhibit MV fusion [20] or a small molecule that potently blocks activity of the MV RNA-dependent RNA polymerase, L-protein complex in vitro [28] have been reported.

The extract as well as a purified component derived from it, (6,6′-hydroxythiobinupharidine) are pleiotropic in their action. Regarding the extract, it has anti-inflammatory activity by downregulating NF-κB [15] and partially preventing LPS-induced septic shock and peritonitis [6]. It is also effective against free as well as intracellular (macrophage) *Leishmania major* parasites [10,11]. The extract has anti-metastatic properties, synergistically with conventional chemotherapy drugs, both in vitro and in vivo. It also induces phospho- extracellular signal-regulated kinases (ERK) expression [14].

We have recently published that purified 6, 6′-hydroxythiobinupharidine, very efficiently and covalently, inhibits human type II topoisomerase [29]. We have preliminary, unpublished data, showing that both the extract and the purified molecule inhibit cathespsins, which are cysteine proteases, likely by forming a disulfide bond at the enzyme’s active site. Lastly, we recently published that the purified molecule primes neutrophils against bacteria present in gum inflammation, enhances phagocytosis, ROS production, and NET formation [30].

We hypothesize that the mechanism of action of nupharidines goes via the electrophilic thiaspirane warhead of nupharidine(s), which targets a nucleophilic cysteine at the active site of cysteine proteases and potentially other enzymes. This mechanism may explain the pleiotropic effect of this family of compounds [31]. The potential medicinal properties of *N. lutea* require specific defined therapeutic windows for each system to be effective. Being pleiotropic and not limited to a single molecular target may be advantageous. By affecting a variety of targets, the probability of developing drug resistance may be reduced.

To better understand the spectrum of activity on NUP, we have tested the extract on the Respiratory Syncytial Virus (RSV), which belongs to the same paramyxoviruses family. Preliminary results show that it is similarly active against the measles virus. NUP is also effective against wild-type MV strains (in addition to the vaccine Edmonston strain) (Table 1). Other viral families will be tested in the future to determine specificity. Further investigation is needed to determine whether NUP’s anti-MV activity is due to its effect on cellular and/or viral targets.

Based on the need to develop new anti-viral drugs, in this study, we present proof of concept that a semi-purified extract of *N. lutea* has anti-viral activity. To further confirm its effectiveness against the measles virus, we will infect transgenic mice with the MV in vivo by infecting transgenic mice expressing the CD46 MV receptor and treating the infected mice with NUP and/or the purified molecule [32]. Effective toxicity experiments should be performed before a human clinical trial can be designed.

In conclusion, we suggest that NUP acts against viral proteins, most efficiently through the P- protein, which is a novel anti-viral target. It is of interest to further establish the mechanism(s) by which NUP affects MV protein expression. Experiments will be planned to determine whether NUP inhibits general viral protein translation or, rather, it induces selective protein degradation (i.e., by ubiquitination). In vivo models of viral infection and treatment are required to demonstrate its potential therapeutic benefit.

## 4. Materials and Methods

### 4.1. Cell Lines and Viruses

L428 cells: Parental L428 cells were derived from a Hodgkin’s lymphoma patient (German Collection of Animal Cultures, Braunschweig, Germany, DSM ACC 197). The cells express both MV receptors CD46 and CD150 (SLAM) [33]. These cells were maintained in RPMI 1640 medium, supplemented with FCS 10%, 1% L-glutamine, and 1% Pen-Strep (Beit Haemek, Israel).

MV persistently infected cells: L428 cells persistently infected with the MV Edmonston strain (L428 + MV) were kindly provided by Dr. Jindrich Cinatl, Institute of Medical and Experimental Virology, Frankfurt/Main Germany.

L428 cells were also persistently infected in our laboratory [34] with the wild type MV strain containing the Green Fluorescent Protein (GFP) gene (IC323-GFP MV), kindly provided by Dr. Yusuke Yanagi, Department of Virology, Faculty of Medicine, Kyushu University Japan).

VERO cells: African green monkey kidney VERO cells are used as hosts to the MV Edmonston strain. They were maintained in Dulbecco’s Modified Eagle’s Medium (DMEM) 10% Fetal Calf Serum (FCS,) 1% L-glutamine, 1% Pen-Strep, and passaged with trypsin EDTA. VERO cells were transfected with the CD150 (SLAM) gene (VERO-SLAM cells) and infected with the IC323-GFP MV (the cells were also provided by Dr. Yusuke Yanagi). The IC323-GFP virus is recognized by the MV receptor CD150 (SLAM) but not by CD46 [35]. The transfected VERO-SLAM cells were maintained in 500 μg/mL G418.

### 4.2. Nuphar Lutea Extracts (NUP)

Floating and submerged leaves of Nuphar lutea were collected from a cultivation pond at the Jacob Blaustein Institutes for Desert Research campus (BIDR), which was started with rhizomes harvested from the Yarkon River in Israel. The specimen voucher-Nuphar169 is a sample maintained at BIDR. The species was identified by Dr. Moshe Agami, the former director of the Botanical gardens of Tel Aviv University.

Preparation of NUP extracts have been described in detail [15]. *Nuphar lutea* L. leaves were oven-dried at 70 °C, and ground in a mortar. Ten grams of dry powder was extracted in 100 mL methanol. The mixed slurry was centrifuged (4000 rpm, 4 °C, 30 min). The supernatant was evaporated under reduced pressure. The residue was dissolved in 100 mL of a mixture of 1N HCl and chloroform (1:1, *v*/*v*). The mixture was separated on a separatory funnel. The aqueous phase was collected and adjusted to pH 9 by adding 25% NH_4_OH. The precipitate was harvested by centrifugation and dissolved in acidic methanol. The solution was placed on a silica gel column that was developed with a mixture of chloroform/ethyl-acetate/diethylamine (20:1:1, *v*/*v*/*v*). The fractions’ activity were monitored using the NF-κB luciferase reporter gene assay [15]. They were then combined and evaporated to dryness under reduced pressure. The residue was dissolved in 50% methanol in water (NUP) and was used in this work.

Partial purification of the active principle and NMR analysis involves the following. The fractions showing NF-κB inhibitory activity were analyzed by one-dimensional and two-dimensional NMR spectroscopy. The ^1^H and ^13^C NMR spectra indicated the presence of several dimeric thioalkaloids. More specifically, typical 1 H-1 H spin systems comprising signals at 7.5–7.2 ppm and 6.6–6.2 ppm were identified by COSY experiments ([15], Supplementary Figure S1). Earlier, H-14 and H-13 of the furan ring systems in thionupharidines and thionuphlutines have been shown to display this characteristic signal pattern [36]. The tentative assignments were further confirmed by Heteronuclear Multiple-Quantum Correlation (HMQC) and Heteronuclear multiple bond Correlation (HMBC) experiments, which displayed the expected patterns due to direct or long-range ^1^H-^13^C couplings, respectively ([15], Supplementary Table S2). From the number of signals detected at specific regions for given atoms of the compound family, it can be concluded that the mixture contained several members of the thioalkaloid family, such as 6-hydroxythiobinupharidine, 6-hydroxythiobinuphlutine B, and 6,6′-hydroxythiobinupharidine ([15], and Figure 6).

The semi-purified extracts used belong to different batches with leaves harvested at different times. The activity of the different extracts varies and are normalized according to their ability to inhibit NF-κB, as measured by a luciferase reporter gene. In addition, different assays (Western blots, immunohistochemistry, viral inhibition, etc.) require different optimal concentrations, especially if the incubation time of the cells with the extract varies (the longer the incubation time, the lower the concentration required). In general, in this manuscript as well as in our previous publications, the concentration range is in the low μ/mL and below.

### 4.3. MV Acute Infection with the MV Vaccine Edmonston Strain

Vero cells (8 × 10^3^/well) were seeded in a 96-well plate. Cells were infected with MV (maximum Multiplicity of infection (MOI) was 2.5) and treated in triplicate with previously determined nontoxic concentrations of NUP (0.3 μg/mL), either 24 h before infection, simultaneously, or 24 h after infection. 96 h after infection, cell survival was determined by the tetrazolium-based XTT cell proliferation (viability) assay kit, (2,3-bis-(2-methoxy-4-nitro-5-sulfophenyl)- 2H-tetrazolium-5-carboxanilide) (Beit Haemek, Israel). Experiments were repeated at least three times.

### 4.4. Western Blot Analysis

Whole cells’ protein lysates (6 × 10^6^ cells per sample) were prepared using RIPA lysis buffer (Tris pH 8, 10 mM, NaCl-100 mM, EGTA pH 8, 5 mM, Beta mercaptoethanol-45 mM, NP40-1%, EDTA pH 8, 10 mM, NaF-50 mM, SDS-0.1%). Quantification of protein was made by the Bradford method (Bio-Rad) and was measured by ELISA at 590 nm. Protein extracts (30–50 μg) were separated by 10% SDS PAGE, transferred, and blotted onto a nitrocellulose membrane (Scleicher & Schuel). The membranes were incubated overnight with several primary antibodies.

The following anti-MV mouse monoclonal antibodies were obtained: Anti- P protein from Argene, NY, USA, anti-N protein from Chemicon International Inc (Temecula, CA, USA). Anti-V antibodies were obtained from Dr. Kaoru Takeuchi, Department of Infection Biology, Graduate School of Comprehensive Human Sciences and Institute of Basic Medical Sciences, University of Tsukuba Japan. Anti- β-actin were obtained from Merck-Sigma, USA. Anti- mouse peroxidase-linked IgG was purchased from Jackson Immuno Research (West Grove, PA, USA). Protein bands were detected by chemiluminescence with EZ- electrochemiluminescence-ECL (Beit- Haemek, Israel).

### 4.5. Immunohistochemistry

Persistently infected L428 + MV cells were treated with different concentrations of NUP and incubated for 12 h at 37 °C. Then cells were cyto-centrifuged at 900 rpm for 5 min and fixed in 4% formalin overnight. Slides were stained with anti-P protein and N protein primary antibodies (the same antibodies used in the Western blot analysis). This was followed by anti-mouse IgG-peroxidase and detected by the ABC-Vectastin immunoperoxidase method (Vector laboratories CA USA) (positive staining is brown). The nuclei were counterstained with hematoxylin (blue).

### 4.6. Infective Virus Release (PFU/mL) from NUP-Treated Cells

VERO-SLAM cells (5 × 10^4^/mL) were seeded in triplicate, in 24-well plates. The cells were infected with the wild type MV IC323-GFP strain (Multiplicity of Infection (MOI)-0.68). The following NUP treatment combinations were tested: (1) No NUP, treatment with vehicle only, (2) 24-h pretreatment with NUP followed by viral infection, (3) Infection with the virus and, 24 h later, incubation with the non-toxic concentration of NUP (0.3 µg/mL). The cells were incubated at 37 °C and supernatants were collected 96-h post-infection. Ten-fold dilutions of the supernatants were added to 8 × 10^4^ VERO-SLAM cells seeded in 96 wells. Six well replicates per treatment in one individual plate. Viral infection in each well was determined by GFP fluorescence under a fluorescent microscope. The virus quantification was determined by the TCID50 method. A positive viral presence was defined as the detection of GFP viral fluorescence in the highest dilution of the supernatant in 50% of the replicate wells (at least three out of six wells). Virus titer was calculated using a standard immunofluorescence assay according to the following calculation [37].

Virus titer = Infectious units X inverse of dilution of supernatant/Volume of inoculum

### 4.7. MV RNA Determination by Real-Time RT-PCR (qRT-PCR)

Total RNA was isolated from 6 × 10^6^ L428 + MV and control L428 cells incubated with different concentration of NUP for 12 h. RNA was purified with the EZ-RNA 2 kit (Beit- Haemek, Israel). RNA concentration and purity were determined by Nano Drop spectrometry 260/280 nm.

qRT-PCR was performed with the AgPath-ID^TM^ one-step kit (Applied Biosystems-AB, USA), which contains the reverse transcriptase and the Taq- man polymerase enzymes. Primers and probes were used against the MV-N and -P cDNA.

Primers were used against N (Metabion, Planegg Steinkirchen, Germany):

F: 5′-TCA GTA GAG CGG TTG GAC CC-3′ (final concentration per sample 125 nM)

R: 5′-GGC CCG GTT TCT CTG TAG CT -3′ (final concentration per sample 250 nM)

Primers were used against P (Metabion, Planegg Steinkirchen, Germany):

F: 5′-AGC TCA GCC GTC GGG TTT-3′ (final concentration per sample 125 nM)

R: 5′-CCT CTA GCC GGC TGG ATT TT-3′ (final concentration per sample 250 nM)

Probe against N (IDT- Integrated DNA Technologies, Coralville, USA).

5′-56-ROX- CAA ACA GAG TCG AGG AGA AGC CAG GGA- 3BHQ 2-3′ (final concentration per sample 75 nM)

Probe used against P (from Applied Biosystems, USA):

5′-VIC-CCGGCCCTGCATC-3′ (final concentration per sample 75 nM)

For normalization, we used the primers and probes of both β-actin and PHP (Panhypopituitarism) genes.

For β-actin, we used Taq-Man Gene Expression Master Mix kit (Applied Biosystems, USA):

F: 5′-AAATCTGGCACCACACCTTC-3′ final concentration – 900 nM

R: 5′-GGGGTGTTGAAGGTCTCAAA-3′, final concentration – 900 nM

Final probe concentration was 250 nM.

PHP primers were obtained from Molbiol, Planegg Steinkirchen, Germany:

F: 5′-CAT GGG AAG CAA GGG AAC TAA TG-3′ (final concentration per sample 900 nM)

R: 5′-CCC AGC GAG CAA TAC AGA ATT T-3′ (final concentration per sample 900 nM)

Probe-5′-Cy5 TCT TCC CTC GAA CCT GCA CCA TCA AT-3′ (final concentration per sample 225 nM)

The final concentration of samples was 50 ng/μL (stock). Samples were diluted 1:100 in DEPC water. From this dilution, 2 µL of RNA were used for qRT-PCR analysis. The final volume of the sample was 20 µL.

Real-time-RT-PCR program:

Step 1-30 min in 50 °C → 10 min in 95 °C → 15 sec in 95 °C

Step 2-32 sec in 55 °C for every cycle, for a total of 40 cycles.

The results were normalized to β-actin and PHP of the control untreated L428 +MV cells.

The experiments were repeated two times.

### 4.8. Statistical Analysis

All experiments were repeated at least three times, unless otherwise stated. Data is presented as an average and standard deviation. Data was analyzed by the one-tailed, two-sample equal variance (homoscedastic) Student’s *t*-test in Microsoft Excel, 2013 Student and Teacher Edition. A *p* value equal to or smaller than 0.05 was considered significant.

## Figures and Tables

**Figure 1 molecules-25-01657-f001:**
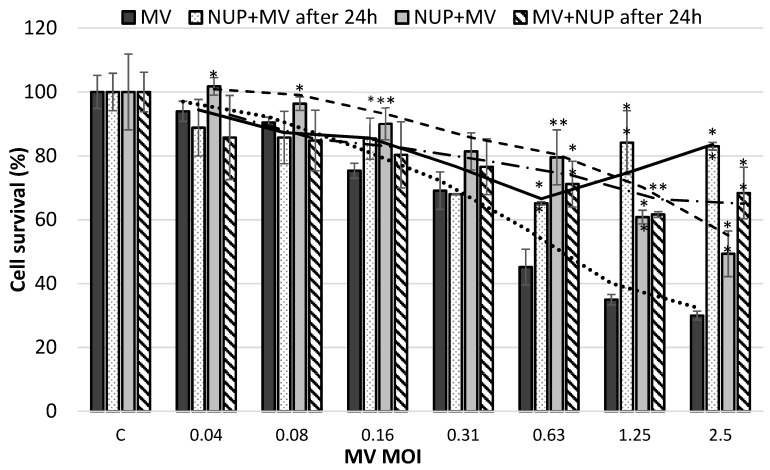
The effect of *Nuphar lutea* (NUP) on survival of measles virus (MV) acutely infected cells. VERO cells were infected with double dilutions of measles virus (MV) (maximum Multiplicity of Infection (MOI)—2.5). The cells were treated with 0.3 µg/mL NUP, either 24 h before infection (dots), simultaneously (light grey), or 24 h after infection (stripes). Cell survival was determined by the tetrazolium based cell proliferation (viability) test 96 h post infection. Statistical analysis (*p* < 0.05 *, *p* < 0.01 **) and a moving average trend line is presented. MV infected but *Nuphar lutea* (NUP) untreated cells were used as a reference for statistical analysis.

**Figure 2 molecules-25-01657-f002:**
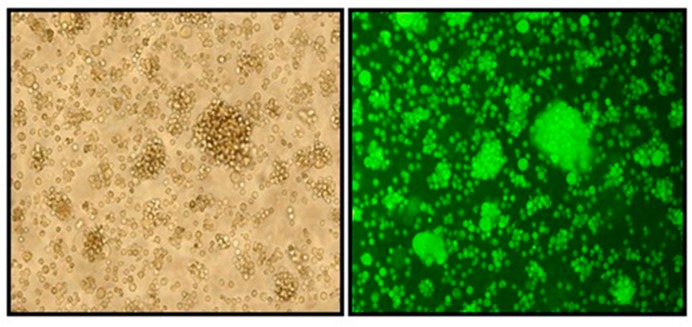
L428 cells persistently infected with MV-GFP. The GFP MV virus was detected by fluorescence microscopy (X 200).

**Figure 3 molecules-25-01657-f003:**
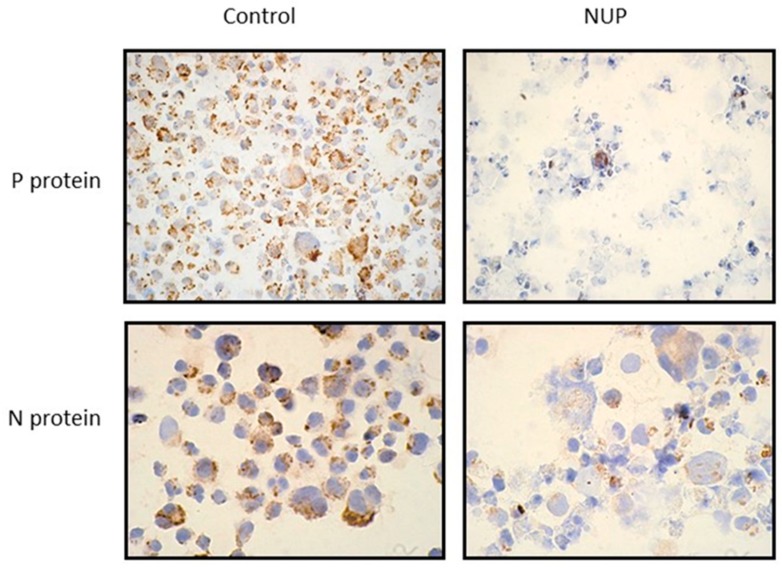
Expression of P- and N-MV proteins following NUP treatment. L428 + MV cells were incubated for 12 h without NUP (Control) or with 6 µg/mL NUP. Immunohistochemistry with antibodies against P- and N-MV proteins was performed (X 400). Positive staining is brown. The nuclei were counterstained with hematoxylin (blue).

**Figure 4 molecules-25-01657-f004:**
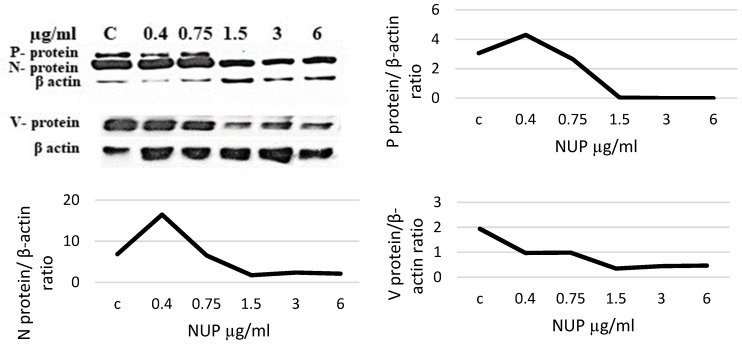
Expression of P-, N-, and V-MV proteins following NUP treatment by a Western blot. Western blot analysis was performed on whole cell lysates from L428 + MV cells. Antibodies used were against P-, N-, and V-MV proteins. L428 + MV cells were incubated with different concentrations of NUP for 12 h. Densitometry of the blot is presented and normalized to β-actin.

**Figure 5 molecules-25-01657-f005:**
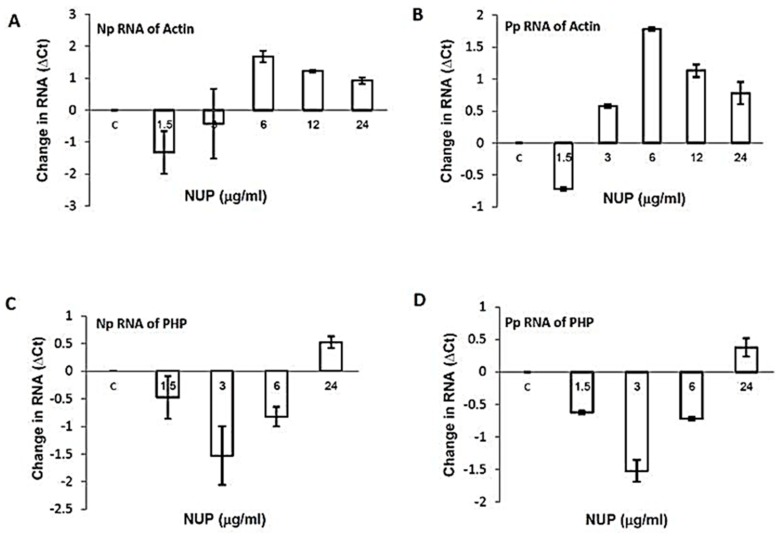
Effect of NUP on P- and N-MV genes’ RNA expression. L428 + MV cells were incubated with different concentrations of NUP for 12 h. After incubation, total RNA isolation and qRT-PCR were performed. MV primers and probes were against N- and P- protein, human β-actin, and PHP genes were used for normalization. After normalization of N- or P- cycles with either β-actin or PHP, the control values were subtracted from the values of the normalized NUP-treated samples (∆Ct). (**A**) N gene normalized to β-actin gene, (**B**) P- gene normalized to β-actin gene, (**C**) N- gene normalized to PHP gene, and (**D**) P-gene normalized to PHP gene.

**Figure 6 molecules-25-01657-f006:**
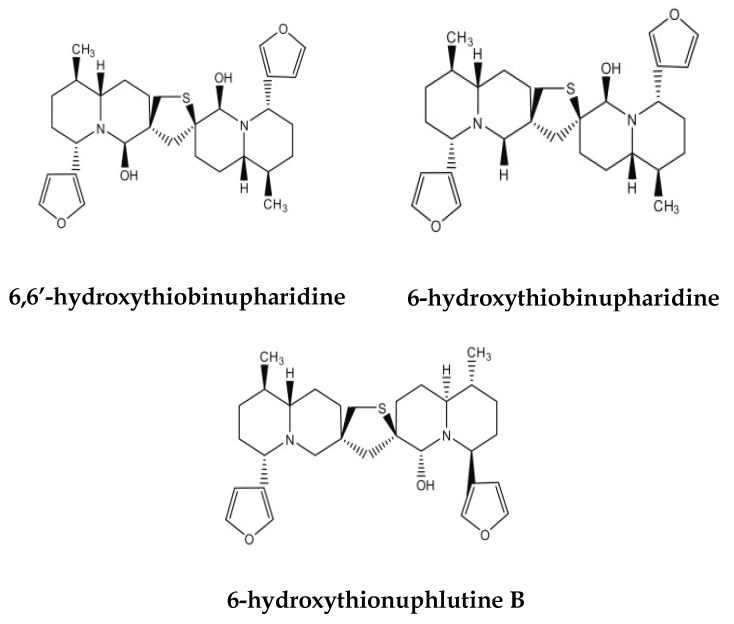
Members of the thioalkaloid family present in the NUP semi-purified extract.

**Table 1 molecules-25-01657-t001:** Effect of *Nuphar lutea* (NUP) on the release of viral particles from measles virus (MV) infected cells.

Samples	MV Titer 96 h after Infection (PFU/mL)
Control MV-infected culture	1.8 × 10^8^
NUP treatment 24 h before infection	1.02 × 10^4^
NUP treatment 24 h after infection	6.6 × 10^5^

The amount of infective virus released to the medium (PFU/mL) in NUP treated (0.3 µg/mL) or untreated cells was titrated. 96 h post infection, supernatants were collected, and ten-fold dilutions were applied to uninfected VERO-SLAM cells. Fluorescence (virus presence) was scored through a fluorescence microscope after 96 h. PFU/mL was determined as described in Materials and Methods.

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
