# Peer review of "Nuphar lutea Extracts Exhibit Anti-Viral Activity against the Measles Virus"

_molecules, 2020, doi:10.3390/molecules25071657_

Round 1

Reviewer 1 Report

Comments

Activity (of any source) against the Measles Virus remains of great current interest, certainly in low-income countries. Yet, the present manuscript does not fulfill a few essential requirements:

  • What is NUP exactly? Just the extract? It needs to be chemically defined, so as to guarantee its reproducibility and confirmation by others.
  • What is the exact mechanism (target) of action?
  • What is the SAR (“structure-activity relationship”)?
  • What is the spectrum of antiviral action (only Measles, or other paramyxoviruses and possibly orthomyxoviruses as well?
  • What are the therapeutic perspectives?

These questions should be addressed in a thoroughly revised version of the present manuscript.

Reviewer 2 Report

This manuscript is describing about anti-viral activity of Nuphar lutea extract against the measles virus. The experiments were organized well and the manuscript was written well. However, it must be revised to improve the descriptions at several points; 1) effective concentration of the extract should be clearly and discussed with other reports, 2) effective compound should be illustrated, at least, the fraction. 3) Abbreviations and full spells should be checked and revised.

Specific Comments;  

  • L17, Nuphar lutea L (NUP, yellow ...)
  • Figure legends should be positions at the bottom of the figures
  • L78, XIT -> full spell
  • Fig. 2 -> MOI -> full spell
  • 24hrs -> 24 h (* Space should be given between numeral and unit for all the data (ex. 24h -> 24 h, 10mM -> 10 mM, ........) 
  • L209-210, should be rewritten.

Round 2

Reviewer 2 Report

All the comments were addressed and it is recommeded to be acceptable to This Journal.